# Probing structural constraints of negation in Pretrained Language Models

**David Kletz**[1,2]  and  **Marie Candito**[1]  and  **Pascal Amsili**[2]

(1) Université Paris Cité & LLF (CNRS/UPC)

(2) Université Sorbonne Nouvelle & Lattice (CNRS/ENS-PSL/USN)

david.kletz@sorbonne-nouvelle.fr, marie.candito@u-paris.fr, pascal.amsili@ens.fr

## Abstract

Contradictory results about the encoding of the semantic impact of negation in pretrained language models (PLMs) have been drawn recently (e.g. Kassner and Schütze (2020); Gubelmann and Handschuh (2022)). In this paper we focus rather on the way PLMs encode negation and its formal impact, through the phenomenon of the Negative Polarity Item (NPI) licensing in English. More precisely, we use probes to identify which contextual representations best encode 1) the presence of negation in a sentence, and 2) the polarity of a neighboring masked polarity item. We find that contextual representations of tokens inside the negation scope do allow for (i) a better prediction of the presence of *not* compared to those outside the scope and (ii) a better prediction of the right polarity of a masked polarity item licensed by *not*, although the magnitude of the difference varies from PLM to PLM. Importantly, in both cases the trend holds even when controlling for distance to *not*. This tends to indicate that the embeddings of these models do reflect the notion of negation scope, and do encode the impact of negation on NPI licensing. Yet, further control experiments reveal that the presence of other lexical items is also better captured when using the contextual representation of a token within the same syntactic clause than outside from it, suggesting that PLMs simply capture the more general notion of syntactic clause.

## 1 Introduction

Negation has recently been the focus of various works aiming at determining the abilities of Pretrained Language Models (PLMs) to capture linguistic knowledge.

Some works investigate the 'semantic impact' of negation, namely its impact in terms of truth values, by interpreting how the presence of negation impacts the probability distribution at a masked position. The rationale is that negating a verb reverses the truth value of its clause, which should be reflected in the probability distribution at certain positions. Ettinger (2020); Kassner and Schütze (2020) use factual statements such as (1), and report that models output similar distributions for the positive and negative variants of (1), and conclude that models largely ignore negation.

(1)     *A robin is (not) a [MASK]*

Gubelmann and Handschuh (2022) chose to avoid factual statements and to focus rather on multi-sentence self-contained examples, such that, given the context provided by the first sentence, one particular word is either likely (in positive items) or ruled out (in negative items) at a masked position in the second sentence. Because this particular word is substantially less often the top-1 prediction in the negative items than in the positive items, the authors draw the opposite conclusion that PLMs do show sensitivity to negation.

A different line of works focused on finding out to what extent negation is encoded in PLM embeddings. Celikkanat et al. (2020) train classifiers taking as input the contextual embedding of a verb or its subject or direct object, and predicting whether the verb is negated or not. The resulting high accuracy allows them to conclude that these tokens' embeddings do contain "traces" of *not*. More generally, several authors have investigated whether the contextual representation of a token encodes information about surrounding tokens. To ease further reading, we will talk of a classifier taking as input an **input embedding**, i.e. the contextual

representation of an **input token**, and predicting some **target information** about another token in the sentence. For instance, Klafka and Ettinger (2020) study how input embeddings encode animacy, gender, and number of surrounding words in a specific SVO context. Li et al. (2022) target the number feature of French participles in the context of object-past participle agreement. They show that the performance of the classifier depends on the syntactic position of the input token in the sentence. We will build on their idea to compare performance at predicting target information depending on the syntactic zone the input token belongs to. In this paper, one of the probed target information will be the presence or absence of a given word within the sentence, which we call the **target token**.

More precisely, our aim is to study PLMs' ability to capture and encode structural information concerning negation (namely negation scope). To do so we first probe whether input embeddings can serve to accurately predict the presence or absence of a target *not*.[1] Moreover, we wish to test PLMs' ability to actually *mobilize* this encoding to capture phenomena that are direct consequences of the presence of negation. To do so, we focus on the licensing of Negative Polarity Items (NPI) by *not* modifying a verb. Polarity Items (PI), either positive (e.g. *some*), or negative (e.g. *any*), are words or expressions that are constrained in their distribution (Homer, 2020). A NPI will require that a word or a construction, called the **licensor**, be in the vicinity. More precisely, the licensor itself grammatically defines a zone of the sentence, called the **licensing scope**, in which the NPI can appear. The adverb *not* modifying a verb is one such licensor. While *any* is licensed by negation in (2-a) *vs.* (2-b), it is not licensed in (2-c), even though the verb is negated, arguably because it is not in the licensing scope[2].

(2)  a.  Sam didn't find any books.
     b.  *Sam found any books.
     c.  *Any book was not found by Sam.

Jumelet and Hupkes (2018) have shown that LSTM embeddings do encode the notion of licensing scope (given an input embedding, a clas-

sifier can predict the structural zone the input token belongs to), a finding later confirmed for transformer-based PLMs (Warstadt et al., 2019). Focusing on when the licensor is a verb-modifying *not*, we rather investigate whether this encoding of the zones *go as far as* enabling a better prediction of a PI's polarity from *inside* the licensing scope compared to *outside* the scope. So instead of the question *"Is this input embedding the embedding of a token located within, before or after the licensing scope?"*, we rather ask the question *"Given a masked PI position, and an input embedding of a neighboring token, what is the polarity of the PI?"*, and we study whether this question is better answered when the input embedding is inside or outside the licensing or negation scopes.

Note that our methodology differs from that of Jumelet and Hupkes (2018), who, given an input token, predict the zone this token belongs to. We instead predict the polarity of a neighboring masked polarity item and then compare accuracies depending on the input token's zone. Our motivation is that the polarity, being a lexical information, requires less linguistic preconception, and hence our probing method is a more direct translation of the NPI licensing phenomenon: we study whether and where the information of *"which PIs are licit where?"* is encoded, in the context of sentence negation. This method also allows us to better control the confounding factor of distance between the input embedding and the licensor *not*.

In the following, we define the linguistic notions of negation scope and NPI licensing scope in section 2, and show how we actually identified them in English sentences. In section 3, we describe our probing experiments and discuss their results, both for the encoding of *not* (section 3.1), and the encoding of NPI licensing (section 3.2). We then study the more general ability of PLMs to deal with clause boundaries (section 4), and conclude in section 5.

## 2 Defining and identifying scopes

### 2.1 Negation scope

From a linguistic point of view, the scope of a negation cue is the area of the sentence whose propositional content's truth value is reversed by the presence of the cue. While in many cases it is sufficient to use the syntactic structure to recover the scope, in some cases semantics or even prag-

---

[1] We restrict our probing to *not*, which is by far the most frequent negation clue (57% of the occurrences, while the second most frequent, *no*, accounts for 21% of occurrences).

[2] We leave aside the uses of *any* and the like having *free choice* interpretations, as for instance in *"Pick any card"*.

matics come into play.[3] Nevertheless, annotation guidelines usually offer syntactic approximations of negation scope.

To identify the negation scope for *not*[4] modifying a verb, we followed the syntactic constraints that can be inferred from the guidelines of Morante and Blanco (2012). Note though that these guidelines restrict the annotation to factual eventualities, leaving aside e.g. negated future verbs. We did not retain such a restriction, hence our identification of the negation scope is independent from verb tense or modality.

## 2.2 NPI licensing scope

Polarity items are a notoriously complex phenomenon. To identify the NPI licensing scope, we focus on specific syntactic patterns defined by Jumelet and Hupkes (2018), retaining only those involving *not* as licensor.[5] Table 1 shows an example for each retained pattern (hereafter the **neg-patterns**), with the NPI licensing scope in blue.

Importantly, in the neg-patterns, the licensing scope is strictly included in the negation scope: within the clause of the negated verb, the tokens to its left belong to the negation scope but not to the licensing scope. E.g. in (3), *anyone* is not licit as a subject of *going*, whether the location argument is itself a plain PP, a NPI or a PPI (3-b).

(3)  a.  I'm not going anywhere.
     b.  *Anyone is not going to the party/ somewhere/anywhere.

We thus defined 4 zones for the *not*+NPI sentences, exemplified in Table 1: **PRE** (tokens before both scopes), **PRE-IN** (to the left of the licensing scope, but within the negation scope), **IN** (in both scopes), and **POST** (after both scopes).

We note though that the restriction exemplified in (3-b) only holds for non-embedded NPIs (de Swart, 1998), so examples like (4), with an embedded NPI in the subject of the negated verb (hence belonging to our **PRE-IN** zone), are theoretically possible.

(4)  Examples with any relevance to that issue didn't come up in the discussion.

Yet in practice, we found that they are extremely rare: using the Corpus of Contemporary American English (COCA, Davies 2015)[6], we extracted sentences matching one of the neg-patterns, and among these, sentences having *any* or *any-body/one/thing/time/where* in the IN zone, the PRE-IN zone or both. As shown in Table 2, *any\** in the PRE-IN zone are way rarer than in the classical licensing scope (IN zone)[7]. Hence we sticked to the usual notion of direct NPI licensing scope, as illustrated in Table 1.

## 2.3 Building the not+NPI test set

Having defined these structural zones, we could use them to probe the traces they carry and compare the magnitude of these traces across the four zones. To do so, we built a test set of COCA sentences containing *not* licensing a NPI (hereafter the **not+NPI** test set), matching one of the neg-patterns of Table 1, and having at least one *any, anybody, anyone, anything, anytime* or *anywhere* within the licensing scope.

The scope of negation has been implemented through an approximation using dependency parses (from the Stanza parser (Qi et al., 2020)), which proved more convenient than phrase-structure parses: we took the subtree of the negated verb, excluding *not* itself, and excluding dependents corresponding to sentential or verbal conjuncts and to sentential parentheticals.

More precisely, we identified the token having *not* as dependent (which, given our patterns, can be either the negated verb or a predicative adjective in case of a negated copula). Then, we retrieved the children of this head, except those attached to it with a "conj", "parataxis", "mark" or "discourse" dependency. In the complete subtrees of the selected dependents, all tokens were annotated as being inside the negation scope.

---

[3]For instance in *Kim did not go to the party because Bob was there.*, negation may scope only over the matrix clause or include the causal subordinate clause.

[4]In all this article, *not* stands for either *not* or *n't*.

[5]We ignored pattern 4 (*never* instead of *not* as licensor), and 6 (too few occurrences in our data). We merged patterns 1 and 2, and corrected an obvious minor error in pattern 5.

[6]We used a version with texts from 1990 to 2012. COCA is distributed with some tokens in some sentences voluntarily masked, varying across distributions. We ignored such sentences.

[7]More precisely, the figures in Table 2 correspond to an upper bound, because of (i) potential syntactic parsing errors impacting the identification of the zones, (ii) cases in which the NPI licensor is different from the *not* targeted by the patterns, and (iii) cases in which *any\** is a free choice item rather than a NPI. We inspected 250 examples of *any\** in the PRE-IN zone, and 250 examples in the IN zone. In the former, we found that almost all cases fall under (i), (ii) or (iii), less than 3% corresponding to examples such as (4)). In contrast, in the IN zone the proportion of NPIs actually licensed by the target *not* is 92%.

| Id | Pattern | Example and zones |
|---|---|---|
| 1/2 | (VP (VB*/MD) ( RB not ) VP ) | I have my taxi and \| I \| 'm not \| going anywhere \| but my brother will leave Spain because he has a degree. |
| 3 | (VP (VB*) RB not ) NP/PP/ADJP ) | Since it is kind of this fairy-tale land, \| there \| aren't \| any rules of logic \| so you can do anything, she says. |
| 5* | (S ( RB not ) VP ) | I went in early, \| not \| wanting anyone to see me \| and hoping for no line at the counter. |

Table 1: The "**neg-patterns**": patterns adapted from Jumelet and Hupkes (2018), which we used to identify some cases of *not* licensing a NPI and to build the *not*+NPI test set. **Col1**: pattern id in Jumelet and Hupkes (2018). **Col2**: syntactic pattern (defined as a phrase-structure subtree, using the Penn Treebank's annotation scheme), with the licensing scope appearing in blue. **Col3**: examples with colors for the four zones: pink for tokens in the PRE zone (before both scopes), purple for PRE-IN (to the left of the licensing scope, but within the negation scope), blue for IN (within both scopes) and green for POST (after both scopes). The NPI licensor is *not*, and appears in yellow.

| Total | IN | PRE-IN | both |
|---|---|---|---|
| 45,157 | 35,938 | 711 | 58 |

Table 2: Number of sentences from the COCA corpus, matching the neg-patterns of Table 1: **Col1**: total number, **Col2-4**: number having *any\** in the IN zone, the PRE-IN zone, and in both zones respectively.

| with *not* | 2,285,000 |
|---|---|
| ↪ with *NPI* | 143,000 |
| ↪ pattern 1 | 30,896 |
| ↪ pattern 3 | 2,529 |
| ↪ pattern 5 | 1,020 |
| ↪ pattern 6 | < 100 |

Table 3: Statistics of the *not*+NPI test set: number of COCA sentences matching the neg-patterns (cf. Table 1), and having at least one *any\** in the IN zone (licensing scope).

For the licensing scope, we parsed the corpus using the PTB-style parser "Supar Parser"[8] of Zhang et al. (2020), and further retained only the sentences (i) matching at least one of the neg-pattern of Table 1 and (ii) having a NPI within the licensing scope (IN zone, shown in blue in Table 1), resulting in the *not*+NPI test set, whose statistics are provided in Table 3.

## 3 Probing for the scopes

Our objective is to study how a transformer-based PLM (i) encodes the presence of a negation (the "traces" of negation) and (ii) models lexico-syntactic constraints imposed by negation, such as the modeling of a NPI licensing scope. Us-

ing the terminology introduced in section 1, we probe whether input embeddings encode as target information (i) the presence of *not* elsewhere in the sentence, and (ii) the polarity of a masked PI. The former focuses on a plain encoding of negation, whereas the latter focuses on whether the encoding of negation can be mobilized to reflect a property (NPI licensing) that is directly imposed by negation. To investigate whether such an encoding matches linguistic notions of scopes, we contrast results depending on the zone the input token belongs to (among the four zones defined for *not* licensing a NPI, namely PRE, PRE-IN, IN, POST) and its distance to *not*.

We studied four PLMs : BERT-base-case, BERT-large-case (Devlin et al., 2019) and ROBERTA-base and ROBERTA-large (Liu et al., 2019). All our experiments were done with each of these models, and for a given model, each experiment was repeated three times. All the sentences we used for training, tuning and testing were extracted from the COCA corpus.

### 3.1 Probing for the negation scope

In preliminary experiments, we extended Celikkanat et al. (2020)'s study by investigating the traces of *not* in the contextual embedding of all the tokens of a sentence containing *not* (instead of just the verb, subject and object).

#### 3.1.1 Training neg-classifiers

We trained binary classifiers (hereafter the **m-neg-classifiers**, with *m* the name of the studied PLM) taking an input contextual embedding, and predicting the presence or absence of at least one *not* in the sentence. In all our experiments, the PLMs parameters were frozen. We trained 3 classifiers for each of the 4 tested PLMs. To train and

---

[8] https://parser.yzhang.site/en/latest/index.html

evaluate these classifiers, we randomly extracted 40,000 sentences containing exactly one *not*, and 40,000 sentences not containing any *not*. These sentences were BERT- and ROBERTA-tokenized, and for each model, we randomly selected one token in each of these sentences to serve as input token. Among these input tokens, we ignored any token *not*, as well as all PLM tokens associated to a contracted negation: for instance *don't* is BERT-tokenized into *don* + *'* + *t*, and ROBERTA-tokenized into *don'* + *t*. These tokens were ignored since they are too obvious a clue for the presence of a verbal negation. Furthermore, in order to homogenize the handling of negation whether contracted or not, we also set aside any modal or auxiliary that can form a negated contracted form. Hence, in *She did leave*, *She did not leave* or *She didn't leave*, the only candidate input tokens are those for *She* and *leave*[9]. We used 64k sentences for training (**neg-train-sets**), and the remaining 16k for testing (**neg-test-set**).

We provide the obtained accuracies on this neg-test-set in Table 4, which shows that performance is largely above chance. We provide a more detailed analysis of the classifers performance in section 3.2.

| Model | BERT$_b$ | BERT$_l$ | ROB.$_b$ | ROB.$_l$ |
|-------|----------|----------|----------|----------|
| **Accur.** | 74.3 | 73.1 | 72.1 | 76.6 |

Table 4: Accuracies of the neg-classifiers on the neg-test-set for each PLM (averaged over 3 runs).

### 3.1.2 Studying results on the *not*+NPI test set

To probe the negation scope, we then used the *not*+NPI test set (cf. section 2), and compare accuracies in PRE-IN vs. PRE, and in IN vs. POST.

Note though that distance to *not* is also likely to impact the classifiers' accuracy. Indeed, by definition the structural zones obviously correlate with distance to *not*. For instance, a token at distance 3 to the right of *not* is more likely to be in the licensing scope than a token at distance 20. Hence, to study the impact of the input token's zone, we needed to control for distance to the negation clue.

We thus broke down our classifiers' accuracy on the *not*+NPI test set, not only according to the input token's zone, but also according to its relative

position to the negation cue. Table 5 shows an example of *not*+NPI sentence, and the zone and relative position to *not* of each token. The target *not* has position 0, and so do all the PLMs' subword tokens involved in the negation complex, and all preceding modal or auxiliary, to homogenize across PLMs and across contracted/plain negation. By construction, the PRE and PRE-IN zones correspond to negative positions, whereas IN and POST correspond to positive ones.

[9]COCA sentences are tokenized and tagged. We detokenized them before BERT/ROBERTA tokenization, in order to get closer to a standard input.

The break-down by position for ROBERTA-large is shown in Figure 1 (results for other models are in appendix figure 4). Two effects can be observed, for all the 4 PLMs: firstly, there is a general decrease of the accuracy as moving away from *not*, for the four zones. This contrasts with the findings of Klafka and Ettinger (2020), who did not observe a distance effect in their experiments, when probing whether the contextual representation of e.g. a direct object encodes e.g. the animacy of the subject. The decrease is more rapid before *not* than after it, which remains to be explained. It might come from the negation scope being shorter before *not* than after it.

Secondly, when looking at fixed relative distances, there is a slight but consistent effect at almost all positions that the accuracy is higher when the input token is in the negation scope (either PRE-IN or IN), than when it is outside (PRE and POST) (the differences are statistically significant at $p < 0.001$, cf. Appendix B). This tendency is more marked for the PRE *vs.* PRE-IN distinction than for the POST *vs.* IN distinction.

This observation can be summarized by computing the average **accuracy gap**, namely the accuracy differences averaged across positions (the average of the purple minus pink bars, and of blue minus green bars in Figure 3), which provide an average difference when a token is within or outside the negation scope. The average accuracy gaps for the four tested models are given in Table 6. It confirms that input embeddings of tokens inside the negation scope do allow for a slightly better prediction of the presence of *not* than those outside the scope. Note that the average difference is stable across models, whose size does not seem to matter. It shows that the strength of the encoding of *not* in contextual representations matches the linguistic notion of negation scope.

| BERT tokens | I | see | I | don | ' | t | know | anyone | here | , | I | must | leave | . |
|---|---|---|---|---|---|---|---|---|---|---|---|---|---|---|
| Zones | PRE | PRE | PRE-IN | not | not | not | IN | IN | IN | IN | POST | POST | POST | POST |
| Distance | -3 | -2 | -1 | 0 | 0 | 0 | 1 | 2 | 3 | 4 | 5 | 6 | 7 | 8 |

| ROBERTA tokens | I | see | I | don' | t | know | anyone | here | , | I | must | leave | . |
|---|---|---|---|---|---|---|---|---|---|---|---|---|---|
| Zones | PRE | PRE | PRE-IN | not | not | IN | IN | IN | IN | POST | POST | POST | POST |
| Distance | -3 | -2 | -1 | 0 | 0 | 1 | 2 | 3 | 4 | 5 | 6 | 7 | 8 |

Table 5: Example sentence from the *not*+NPI test set: structural zones and relative positions to *not*. Any auxiliary or modal preceding the target *not* has position 0 too, to homogenize contracted and plain negation, and BERT versus ROBERTA's tokenization.

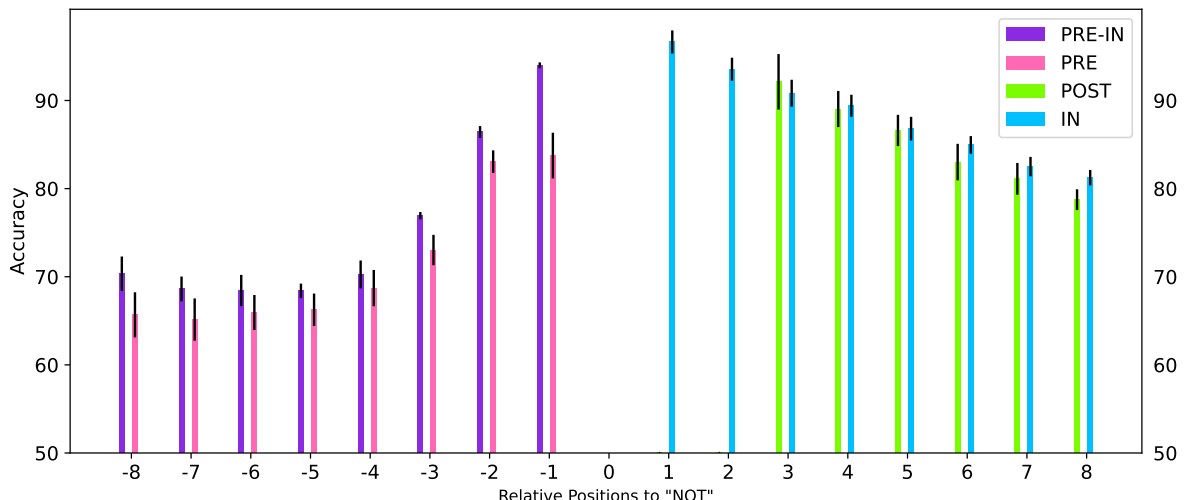

Figure 1: Accuracy of the ROBERTA-large-neg-classifier (average on 3 runs) on the *not*+NPI test set, broken down by zone (colors of the bars) and by relative position to *not* (horizontal axis). Further distances are omitted for clarity. No licensing scope contains less than 2 tokens, hence positions 1 and 2 are always in the IN zone. The bar differences at each position and run are statistically significant at $p < 0.001$ (cf. Appendix B). Figures for the other 3 models are provided in appendix figure 4.

| $BERT_b$ | $BERT_l$ | $ROB_b$ | $ROB_l$ |
|---|---|---|---|
| 3.0 (0.6) | 3.5 (0.2) | 2.6 (0.2) | 2.6 (1.3) |

Table 6: Accuracy gaps for the neg-classifiers on the *not*+NPI test set, for each tested PLM, averaged over 14 relative positions and 3 runs (stdev within brackets).

We also observed that the biggest difference is at position -1, which mostly corresponds to a contrast between a finite *vs.* non-finite negated verb (neg-patterns 1/2/3 *vs.* neg-pattern 5 in Table 1), which seems well reflected in PLMs' embeddings.

### 3.2 Probing for the licensing scope

We then focused on whether this encoding of *not* can actually be **mobilized** to capture the licens-

ing of a NPI. We built classifiers (hereafter the ***m-pol-classifiers***[10], *m* referring to the PLM), taking an input contextual embedding, and predicting as target information the polarity of a masked position, originally filled with a positive or negative PI. Importantly, the input embedding in the training set is randomly chosen in the sentence, and can correspond to a position with no a priori linguistic knowledge about the polarity of the PI (Figure 2).

We train on sentences originally having either a PPI or a NPI, which we mask before running each studied PLM. More precisely, in each COCA sub-corpus (each genre), and for each of the 6 NPI/PPI pairs listed by Jumelet and Hupkes (2018)[11], we randomly took at most 2,000 sentences containing

[10]Full details for all classifiers are provided in Appendix A.
[11]*(any/some)(∅/where/one/body/thing/time)*

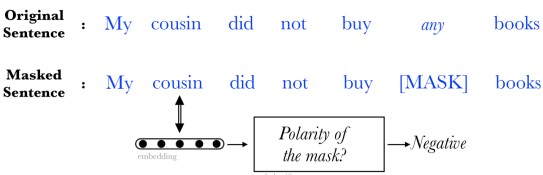

Figure 2: Illustration of the training of the pol-classifiers.

the NPI, and the same amount of sentences containing the corresponding PPI[12]. In each of these, we masked the PI, randomly selected one token per sentence to serve as input token (excluding the masked position) and split these into 63,529 examples for training (**pol-train-set**) and 15,883 for testing (**pol-test-set**).

| Model | BERT$_b$ | BERT$_l$ | ROB.$_b$ | ROB.$_l$ |
|---|---|---|---|---|
| **Accur.** | 64.2 | 63.7 | 56.6 | 68.6 |

Table 7: Accuracies of the pol-classifiers on the pol-test-set for each PLM (averaged over 3 runs).

Accuracies on the pol-test-set for each PLM are shown in Table 7. While still above chance, we observe that it doesn't exceed 69%, which is quite lower than the accuracies of the neg-classifiers (Table 4). This is not surprising since the task is more difficult. First, as stressed above, some of the training input tokens are independent, from the linguistic point of view, of the PI's polarity. Second, the cues for predicting the polarity are diverse. And third, in numerous contexts, both polarities are indeed possible, even though not equally likely. We did not control the training for this, on purpose not to introduce any additional bias in the data. We can thus interpret the pol-classifier's scores as how likely a given polarity is.

Next, we applied these classifiers on the ***not*+NPI** test set. The objective is to compare the classifiers' accuracy depending on the structural zone the input token belongs to. If PLMs have a notion of licensing scope, then the polarity prediction should be higher when using an input token from the IN zone.

---

[12]For *any/some(∅/one/thing)*, we took $2 \times 2000$ occurrences. For *any/some(body/time/where)*, less occurrences were available in some of the subcorpora. We took as many as possible, but keeping a strict balance between NPI and PPI sentences (between $2 \times 169$ and $2 \times 958$ depending on the corpus genre and on the NPI/PPI pair).

### 3.2.1 Results

Once more, we controlled for distance of the input embedding to *not*. The break-down by position and structural zone for ROBERTA-large is provided in Figure 3 (results for other models are in appendix figures 5).

Again, we observe a general accuracy decrease as moving away from *not*, even faster than for the previous experiment. The decrease is more rapid in the PRE-IN zone than in the IN zone (e.g. at distance -4 in PRE-IN, accuracy is less than 70%, whereas it is still above it at distance 8 in the IN zone), which could indicate that the traces of *not* are more *robust* in the licensing scope.

Secondly, as for the previous experiment, for each relative position, **when the input token is in the negation scope (either PRE-IN or IN), the accuracy is higher than when it is outside (PRE and POST)**. Even though we cannot exclude that the relatively high overall accuracies may be explained by the classifier catching some regularities of the sentences containing a NPI rather than a PPI (independently of the presence of *not*), it remains that for the *not*+NPI sentences, accuracy is higher when the input token is in the negation scope than outside it. Moreover, this trend is much more marked than for the previous experiment.

Thirdly, the amplitude of this observation depends on the model. We provide the accuracy gaps for each PLM in Table 8. We observe that the trend is marked for ROBERTA-large and BERT-base (gap of 8.7 and 7.4 accuracy points, actually much higher than the accuracy gaps for predicting the presence of *not*), but lower for ROBERTA-base and BERT-large.

| BERT$_b$ | BERT$_l$ | ROB$_b$ | ROB$_l$ |
|---|---|---|---|
| 7.4 (0.5) | 3.1 (0.4) | 1.4 (0.2) | 8.7 (0.6) |

Table 8: Accuracy gaps for the pol-classifiers on the *not*+NPI test set, averaged over 14 relative positions and 3 runs (stdev within brackets).

This leads us to conclude that (i) PLMs do encode structural constraints imposed by *not* (NPI licensing), but to varying degrees across the PLMs we tested, and (ii) that this encoding is stronger in the negation scope than outside it, independently of the distance to *not*. This only partially matches the linguistic expectation that the strongest zone should be the licensing scope rather than the entire negation scope.

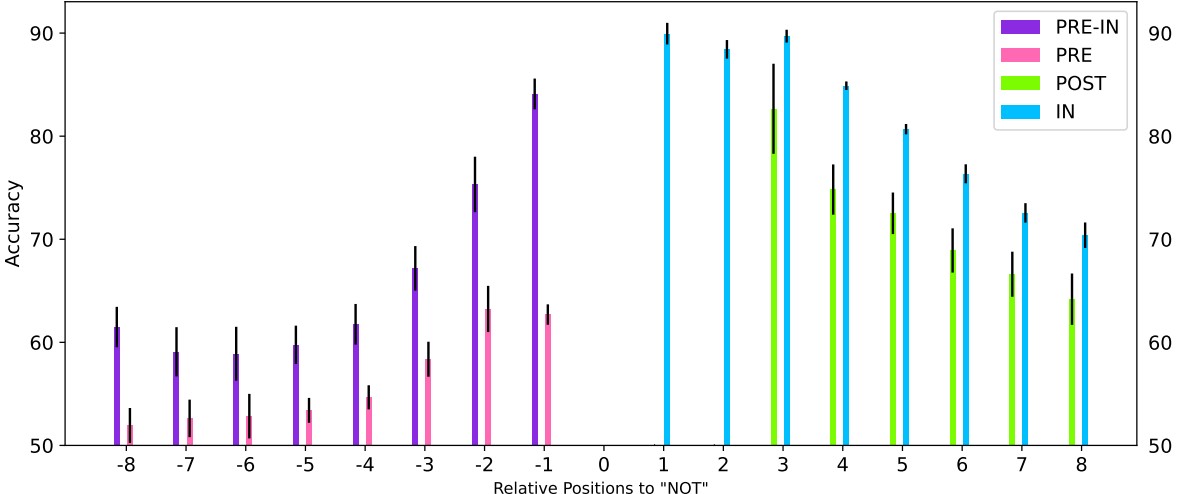

Figure 3: Accuracy of the ROBERTA-large-pol-classifier (average on 3 runs) on the *not*+NPI test set, broken down by zone (colors of the bars) and by relative position to *not* (horizontal axis). Further distances are omitted for clarity. No licensing scope contains less than 2 tokens, hence positions 1 and 2 are always in the IN zone. The bar differences at each position and run are statistically significant at $p < 0.001$ (cf. appendix figures 5).

## 4 Probing clause boundaries

We have seen that PLMs are able to encode negation scope, however this notion of scope often simply corresponds to the notion of syntactic clause. So it might be the case that PLMs are mainly sensitive to clause boundaries and that this sensitivity is the unique/main source of PLMs ability to encode negation scope. In this section we report a number of experiments designed to assess PLMs ability to encode clause boundaries in general.

We chose to use the same setting as the one we used with the neg-classifiers (section 3.1.1). Instead of using *not* as a target token, we chose various tokens with a similar number of occurrences, but other POSs: *often*, *big*, *house*, *wrote*. We trained classifiers to predict whether the target token is in the neighborhood of the input token. This time, the objective is to compare these classifiers' accuracies depending on whether the input token is or isn't in the same clause as the target token (instead of whether the input token is within or outside the negation scope). And just as we did for the neg-classifiers, we will control for distance to the target token by breaking down the accuracies according to the distance between the target and the input tokens.

### 4.1 Training the classifiers with alternative target tokens

To train such classifiers, we repeated the same protocol as for the neg-classifiers: for each target word *often*, *big*, *house*, *wrote*, we randomly selected a balanced number of sentences containing and not containing it, and we randomly picked an input token within each sentence, independently of the presence of the target token, and in case of presence, independently of the clause boundary of the target token. We then split the examples into training (25.5k) and test sets (6.5k). We restricted ourselves to a single PLM, ROBERTA-large. The performances on the training and test sets are provided at Table 9. We note that performance is comparable for all the four target tokens, and comparable to that of the neg-classifiers (cf. Table 4, 76.6 for ROBERTA$_l$): the negation clue *not* is not particularly better encoded in contextual embeddings compared to other open-class target words.

| Target token | house | often | big | wrote |
|:---:|:---:|:---:|:---:|:---:|
| Accur. | 79.1 | 77.1 | 75.2 | 81.2 |

Table 9: Accuracy of the classifiers on test-sets, for the four alternative target tokens, when using ROBERTA-large embeddings (average on 3 runs).

| Target token | In | Out | Accuracy gap |
|---|---|---|---|
| *house* | 83.7 | 79.0 | 4.7 |
| *often* | 82.0 | 76.5 | 5.5 |
| *big* | 80.5 | 79.4 | 1.1 |
| *wrote* | 85.7 | 82.4 | 3.3 |

Table 10: Average accuracy when the input token is within a window of 8 tokens before and 8 tokens after the target token, broken down according to whether the input token is (*In*) or isn't (*Out*) in the same clause as the target token, and accuracy gap (*In* minus *Out*). The results are computed the study-test-set of each target word, using the classifiers trained on ROBERTA-large embeddings.

## 4.2 Studying results when input tokens are within or outside the same clause

In order to study whether PLMs do encode the notion of syntactic clause, we compared the classifiers' performance when the input token is or isn't within the same clause as the target token. For each target word, we built a **study-test-set** of 40,000 COCA sentences containing it. We parsed these sentences, and annotated each of their tokens (1) according to their distance to the target token, and (2) as belonging or not the the same clause as the target token.[13]

As in section 3.1.2, we now define accuracy gaps as the average difference between a classifier accuracy on input tokens that are within the same clause as the target token, minus the accuracy on input tokens from outside the clause. Table 10 shows the average accuracy gaps, for input tokens at distance at most 8 from the target token.

The results show that for the 4 tested target words, predicting the presence of the target token is better achieved using an input token from the same clause than from outside the clause. Interestingly, the gaps are higher when the target token is a noun, verb or adverb, and less pronounced for the adjectival target token. Strikingly, except for the adjective *big*, the observed accuracy gaps are even bigger than that obtained using *not* as target token (cf. 2.6 for ROBERTA$_l$ in Table 6).[14] This

tends to indicate that the encoding of the negation scope observed in section 3.1 stems from a more general encoding clause boundaries.

Moreover, breaking-down the results by relative position to the target token (cf. figures 6 in Appendix), shows that the distance to the target token remains by far the most impactful factor.

## 5 Conclusion

In this paper, we studied the way negation and its scope are encoded in contextual representations of PLMs and to what extent this encoding is used to model NPI licensing.

We trained classifiers to predict the presence of *not* in a sentence given the contextual representation of a random input token. We also trained classifiers to predict the polarity of a masked polar item given the contextual representation of a random token. A test set of sentences was designed with *not* licensing an NPI, inside which we identified the negation scope , and the licensing scope.

For these sentences, we found that the contextual embeddings of tokens within the scope of a negation allow a better prediction of the presence of *not*. These embedding also allow a better prediction of the (negative) polarity of a masked PI. These results hold even when controlling for the distance to *not*. The amplitude of this trend though varies across the four PLMs we tested.

While this tends to indicate that PLMs do encode the notion of negation scope in English, and are able to further use it to capture a syntactic phenomena that depends on the presence of *not* (namely the licensing of a negative polarity item), further experiments tend to show that what is captured is the more general notion of clause boundary. Indeed, negation scope is closely related and often amounts to negation scope. Using alternative target tokens with varied parts-of-speech, we find that classifiers are better able to predict the presence of such target tokens when the input token is within the same syntactic clause than when it is outside from it. These results lead us to conclude that knowledge of the negation scope might simply be a special case of knowledge of clause boundaries. Moreover, distance to the target token is way stronger a factor than the "being in the same clause" factor. We leave for further work the study of other factors, such as the POS of the input token, as well as the study of the differences in amplitudes observed between the PLMs we tested.

---

[13]We identified the clause of the target token as the subtree of the head verb of the target token, in the dependency parse.

[14]The gaps are not strictly comparable though, due for our defining the negation scope as a subset of the clause, filtering out sentential conjuncts and sentential parenthetical, cf. section 2.3.

## Acknowledgements

We thank the reviewers for their valuable comments. This research was partially funded by the Labex EFL (ANR-10-LABX-0083).

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

## A Hyperparameter tuning for the neg-classifiers and the pol-classifiers

The PLMs' contextual representations were obtained using a GeForce RTX 2080 Ti GPU.

The neg-classifiers, the pol-classifiers and the classifiers used to predict the presence of other taget tokens were trained on a CPU, each training taking about 15 minutes. Then, testing them on the *not*+NPI test set took about 5 minutes.

To tune these classifiers, we performed a grid search with: a number of hidden layers included in [1, 2], number of units in each layer in [20, 50, 100 450, 1000], and the learning rate in [1, 0.1, 0.01, 0.001].

We selected a learning rate of 0.001, 2 hidden layers, with size 450 each, based on the accuracies on the neg-test-set and the pol-test-set. Except when the learning rate equaled 1, all hyperparameter combinations resulted in similar performance (less than 1 point of accuracy, in the results of figure 3).

The code and methodology was developed first using the BERT-base model, and then applied to the other models. Including code and methodology development, we estimate that the experiments reported in this paper correspond to a total of 160 hours of GPU computing.

## B Statistical significance test

In this section we detail the test performed to assess the statistical significance of the accuracy differences illustrated in Figures 3 and 5.

For each of the four tested PLMs, and for each of 3 runs of classifier training,

- for each position from -8 to -1 relative to the *not*,
  - we compare the accuracy of the pol-classifier in the PRE-IN zone versus in the PRE zone (i.e. the difference between the purple bar with respect to the pink one).
  - namely, we test the statistical significance of the following positive difference : accuracy for tokens in PRE-IN zone minus accuracy for tokens in the PRE zone.

- for each position from 3 to 8,
  - we test the statistical significance of the following positive difference : accuracy for tokens in IN zone minus accuracy for tokens in the POST zone (i.e. the difference between the blue bar with respect to the green one)

Each test is an approximate Fisher-Pitman permutation test (with 5000 random permutations, performed using the script of Dror et al. (2018), https://github.com/rtmdrr/testSignificanceNLP.git), and all the differences listed above result as statistically significant at $p < 0.001$.

## C Supplementary figures

The break-downs by position for the three models not presented in the main text (BERT-base, BERT-large and ROBERTA-base) are provided in Figures 4 (neg-classifiers) and 5 (pol-classifiers).

The break-downs by position for other target tokens are provided in Figures 6

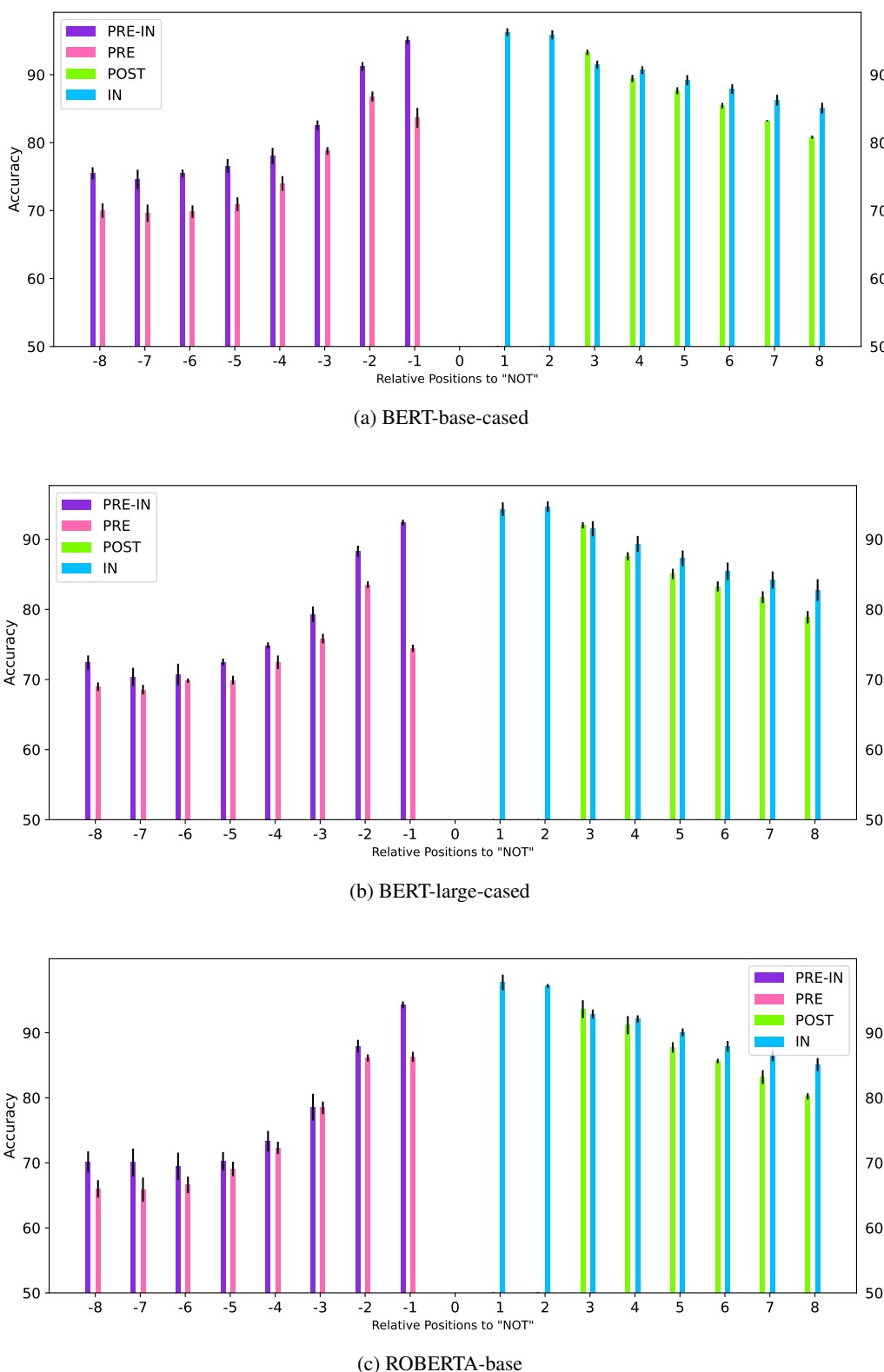

(a) BERT-base-cased

(b) BERT-large-cased

(c) ROBERTA-base

Figure 4: Accuracy (average on 3 runs) of the other neg-classifiers (BERT-base, BERT-large and ROBERTA-base) on the *not*+NPI test set, broken down by zone (colors of the bars) and by relative position to *not* (horizontal axis). Further distances are omitted for clarity. No licensing scope contains less than 2 tokens, hence positions 1 and 2 are always in the IN zone. The bar differences at each position and run are statistically significant at $p < 0.001$ (cf. Appendix B).

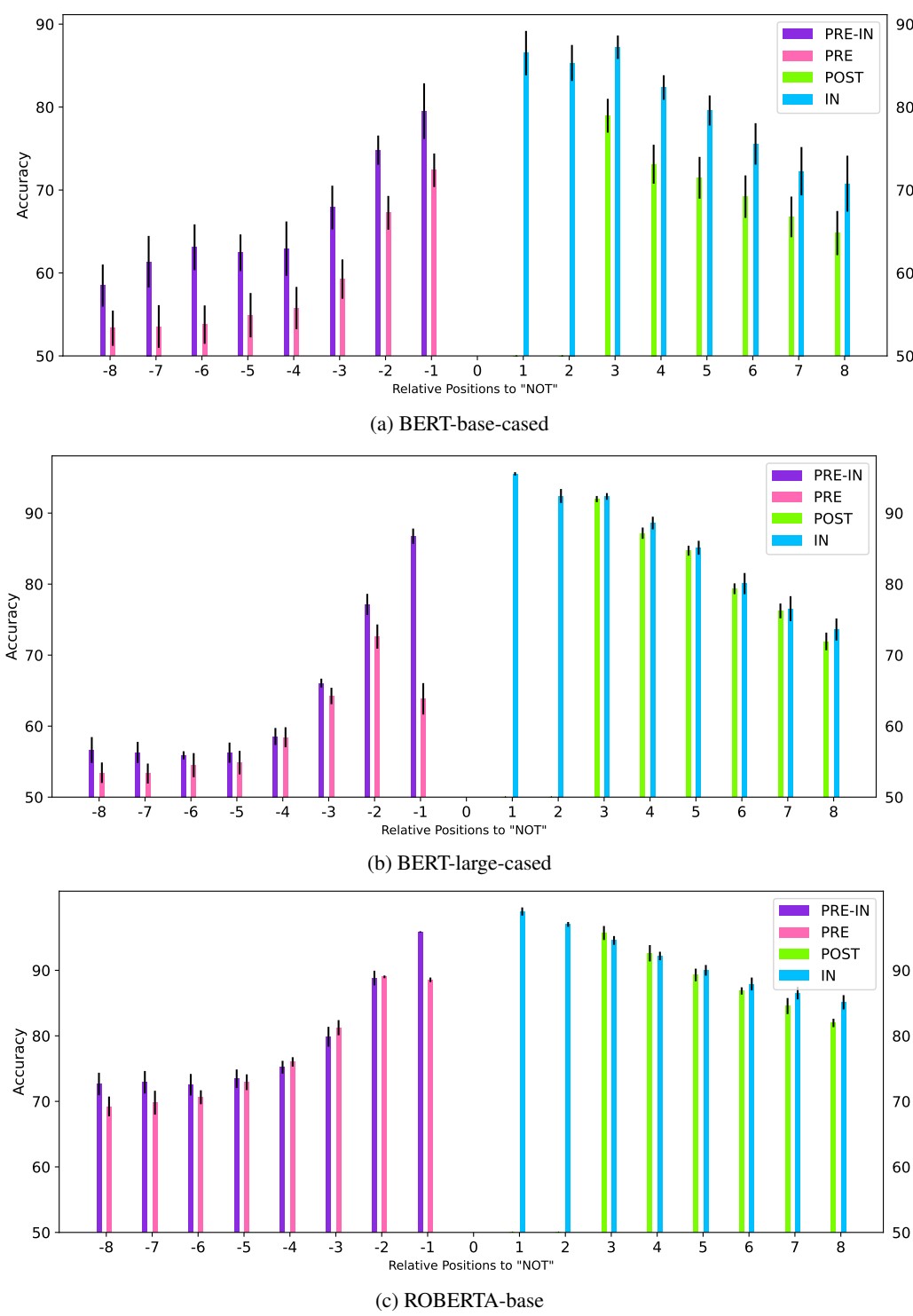

Figure 5: Accuracy (average on 3 runs) of the other pol-classifiers (BERT-base, BERT-large and ROBERTA-base) on the *not*+NPI test set, broken down by zone (colors of the bars) and by relative position to *not* (horizontal axis). Further distances are omitted for clarity. No licensing scope contains less than 2 tokens, hence positions 1 and 2 are always in the IN zone. The bar differences at each position and run are statistically significant at $p < 0.001$ (cf. Appendix B).

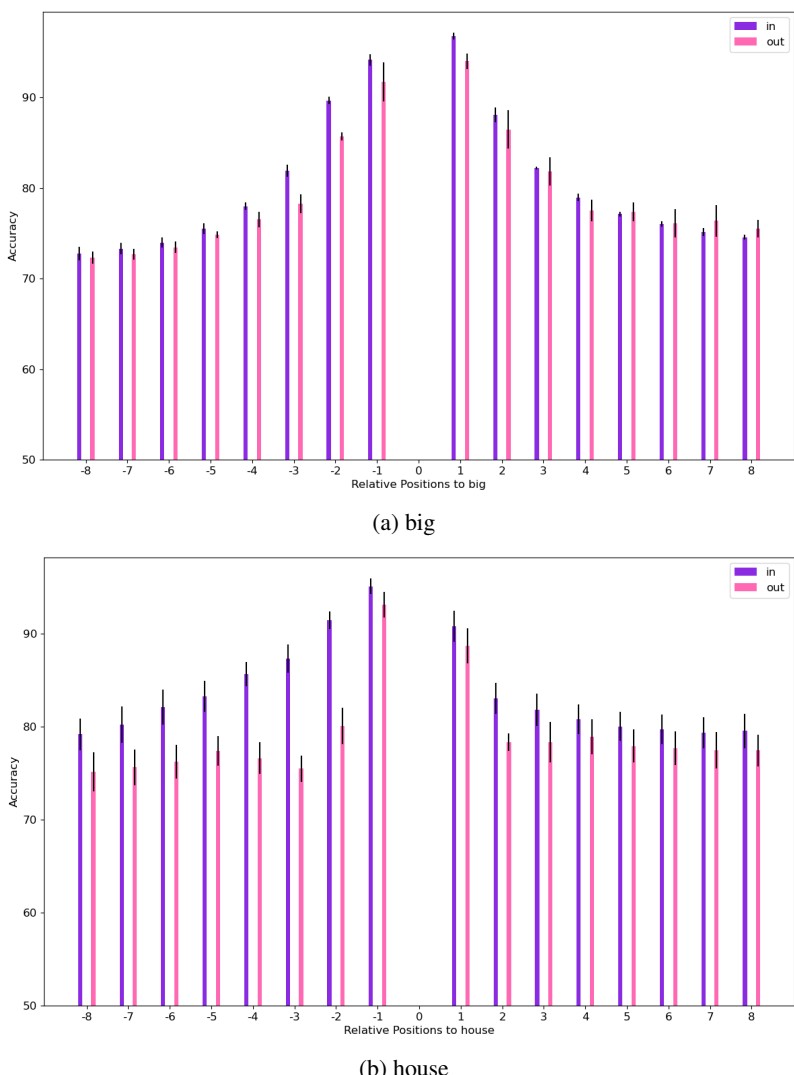

(a) big

(b) house

Figure 6: Accuracy (average on 3 runs) on trace identification tasks. The target tokens are *big* and *house*, and the probed embeddings are from a ROBERTA-large LM. Results are broken down by zone (colors of the bars) and by relative position to *not* (horizontal axis). Further distances are omitted for clarity. The bar differences at each position and run are statistically significant at $p < 0.001$ (cf. Appendix B).