# OpenReview forum: "Probing structural constraints of negation in Pretrained Language Models"
_NoDaLiDa/2023/Conference — NoDaLiDa 2023_

### Official Review · Reviewer_u5b4 · 2023-02-21
**Thorough investigation of the sensitivity of PLMs to NPI licencing constraints**

**Rating:** 8
**Confidence:** 4

**Review:**

The paper addresses a subtle linguistic problem, negative polarity and licencing, and investigates to what extent PLMs are sensitive to structural constraints on licencing of such items. The paper is very detailed and informative. The main conclusion is that the licencing scope of negation for NPIs is marked by PLMs. The methodology for probing uses tokens inside/outside the scope and uses their embeddings to train a classifier for predicting presence of negation and polarity of PI. The methodology is sound and conclusions are backed up by experiments.

Comments

Tables 3 and 4 give a distribution over genres. It is not clear how to interpret these numbers. If the goal is to estimate relative frequency of NPI in context of not, it might be better to give percentages in line 2 of table 3. And if goal is to study whether genre influences percentage of not + NPI in table 4, it might be better to give relative frequencies in table 4. (Although I seem to rember COCA is a balanced corpus with equal portions of each of the genres, in which case relative freq would give the same result as the raw counts as given in the paper.)

It might help to state more explicitly the relationship between 3.1.1. (testing for neg, irrespective of scope constraints) and 3.1.2 (taking scope into account). The way I read this eventually is that table 3.5. gives average performance across context tokens, while fig 1 gives scores per scope position. It might also be informative to give results for 3.1.1. per position as well. I.e. a skeptical reader might observe that while there are accuracy gaps between in and out of scope tokens, in general, distance from NOT is by far the most important factor influencing accuracy. (similar remarks apply to table 8 and fig 3, although the acc gap is more significant there.)









**Paper Type:**

Long paper

---

### Official Review · Reviewer_M4up · 2023-03-09
**Negation in English LMs**

**Rating:** 3
**Confidence:** 3

**Review:**

This paper is a continuation of the line of work that probes a language model's internal representations for evidence whether or not certain linguistic phenomena are captured. Specifically, here the authors probe English LMs to see how they encode the presence of negation and its scope.

The authors extract sentences with negation, specifically "not", and negative polarity items (NPI, e.g., any, anything) and sentences which do not have this. This data is used to train and test probes: one for the presence of "not" and one to predict the polarity of a masked NPI (any/some, etc). The authors compare using the contextualized embedding of a word within negation scope (parametrized in various ways) as a feature for a classifier (it is unclear what classifier is used) and find that the embeddings of words found in negation scope are more predictive of the NPI items.

The paper concludes that LMs do encode a notion of negation scope, but not negative polarity licensing scope.

The topic of the paper is interesting and timely, but currently specific research question and methodology seem unclear. First, I'm not sure what the hypothesis is regarding the probes. If words inside negation scope are more predictive of the negator "not", what does this tell us about LMs and negation? What does this tell us that previous studies do not? This should be made much clearer in the paper. Secondly, the methodology is not described in enough detail: what classifiers were used to probe the models, etc...

I'm also unclear about why using (I suppose) linear classifiers trained on the contextualized embeddings is a more appropriate methodology than using a model's masked-language objective? What do the probes tell us that directly prompting the model does not?

Finally, most of the results are expressed as tendencies that happen at varying degrees depending on the models. However, from these results, the authors make quite strong conclusions. I would perhaps look further into these results and try to find stronger trends or reduce the strength of the claims in the conclusion.

Questions:
---------
- Why do you only use the negator "not"? This seems somewhat limiting, as it is far from the only negation item and the approach used to probe the LMs is general enough to deal with any negator.
- What classifier do you use for your probes? This needs to be explained explicitly and clearly.
- Line 455: you mention that performance is significantly above chance. What significance test did you do to test this?
- Line 513: What do you mean by a slight but almost systematic effect? This sounds like no effect...
- Line 537: How do the results in Table 7 show that the LM matches the linguistic notion of negation scope? This does not seem obvious to me.
- Line 641: What do you mean by the phrase "This avoids using linguistic preconceptions while building the classifiers"? What preconceptions would we run into?


Comments:
---------
- The introduction is currently a mix of introduction and related work. It would perhaps be easier to read if these two were separated out into distinct sections.
- Avoid contractions such as "isn't"


**Paper Type:**

Long paper

---

### Official Review · Reviewer_uu8v · 2023-03-10
**Review: Probing structural constraints of negation in Pretrained Language Models**

**Rating:** 8
**Confidence:** 4

**Review:**

This article studies how negation is encoded in PLM. This is done by selecting a random token in the sentence and querying whether or not it was negated. To further evaluate the results several patterns of negation are considered, and the experiments take into account the distance of the negation in relation to the randomly selected token.

In all, the paper presents and interesting and detailed analysis of the results obtained. The conclusions are presented in a clear manner. It's a clear accept for me.

Some formatting notes:
* Formatting comment: Keep the abstract to one paragraph.
* Formatting comment: Can you make the bars in Figure 1 and 3 (and the same figures in the Appendix) wider? They are a bit difficult to read at the moment.

Content notes:
* 436/437: What do you mean by PLM token? Is this a sub-word embedding, or a word embedding? And if it is word embeddings, what method was used to create them? In general, I think this kind of study needs to consider more carefully what type of representation is probed.
* Do you notice any differences for part-of-speech tags in the sentence/scope? That is, are traces of negation found equally in nouns, verbs, determiners, and so on?
* When the model is trained, are you updating the PLM parameters also? Because if you update the PLM parameters, it seems like you are asking if "representations of negation can be learnt" given that you train the model for this task.
* You mention genres in the description of the data, but no experiments take this into account as far as I can tell. Is this intended?



**Paper Type:**

Long paper

---

### Decision · Program_Chairs · 2023-03-17

Accept